# Enhancement and Validation of a 3D-Printed Solid Target Holder at a Cyclotron Facility in Perth, Australia

**Sun Chan [1,*], David Cryer [1] and Roger I. Price [1,2]**

[1]  Sir Charles Gairdner Hospital, Department of Medical Technology and Physics, Nedlands 6009, Australia; RAPID@health.wa.gov.au

[2]  Department of Physics, University of Western Australia, Nedlands 6009, Australia; roger.price@uwa.edu.au

**\***  Correspondence: sun.chan@health.wa.gov.au

**Abstract:** A 3D-printed metal solid target using additive manufacturing process is a cost-effective production solution to complex and intricate target design. The initial proof-of-concept prototype solid target holder was 3D-printed in cast alloy, Al–7Si–0.6Mg (A357). However, given the relatively low thermal conductivity for A357 ($\kappa_{max}$, 160 W/m·K), replication of the solid target holder in sterling silver (SS925) with higher thermal conductivity ($\kappa_{max}$, 361 W/m·K) was investigated. The SS925 target holder enhances the cooling efficiency of the target design, thus achieving higher target current during irradiation. A validation production of $^{64}Cu$ using the 3D-printed SS925 target holder indicated no loss of enriched $^{64}Ni$ from proton bombardment above 80 μA, at 11.5 MeV.

**Keywords:** cyclotron; targetry; solid target; metal 3D-printing; target temperature; radiometals; radionuclides

## 1. Introduction

We investigated enhancement of the design and material used in the construction of an existing 3D-printed cast alloy (A357) solid target holder for improved thermal management and shuttle movement. The enhanced solid target, constructed in sterling silver, enables the target to withstand the maximum extracted target current from an IBA Cyclone® 18/18 cyclotron at a degraded energy of 11.5 MeV. The improved thermal management of solid target material significantly improved the target yield and/or irradiation time, in order to achieve equivalent isotopic activity. The 'non-slanted' target design ensures compactness and a minimal amount of expensive enriched target material needed for each irradiation. The 3D-printed sterling silver solid target incorporates the experimentally derived thermocouple measurement configuration [1] for continuous temperature monitoring during an irradiation. The cost of the 3D-printed sterling silver target holder, at $650 USD, offers an economical alternative to the traditional subtractive manufacturing technique.

## 2. Materials and Methods

In order to satisfy the complex and intricate design of the cooling mechanism, 3D-printing was used over traditional manufacturing techniques. The sterling silver (SS925) solid target holder was ordered online via the Shapeways 3D-printing manufacturing portal. Due to the inability to directly translate the design for different 3D-printed metals, minor changes were made to the cooling cavity to comply with the material specifications and manufacturing requirements.

The previous 3D-printed target [1] was fabricated from industrial cast alloy (Al–7Si–0.6Mg, A357) on the ReaLizer® SLM (selective laser melting) platform. The target body was annealed at 500

°C for 4 hours, and then furnace-cooled to room temperature to purge the material of excess Si. The theoretical maximum thermal conductivity for cast alloy using this method is 160 W/m·K. Thermal conductivity of sterling silver (SS925) is approximately 2.2 times higher than A357, with a theoretical maximum at 361 W/m·K. The composition of SS925 is predominately silver and copper, 92.5% and 7.5%, respectively. Commercial 3D-printing for sterling silver utilizes a lost-wax process, where a plaster mold of the 3D-printed design is created before molten silver is added into the cast. The 3D-printed holders had minimal post-fabrication treatment with light polishing of accessible surfaces only. In-house machining of the 3D-printed part included O-ring grooves, thermocouple hole, and polishing the interface where the target material is located.

Figures 1 and 2 show the 3D-printed target holder inside the complete solid targetry assembly. The mounting flange for the solid target incorporates a graphite collimator and graphite degrader (dia. 10 mm, thickness 1.0 mm).

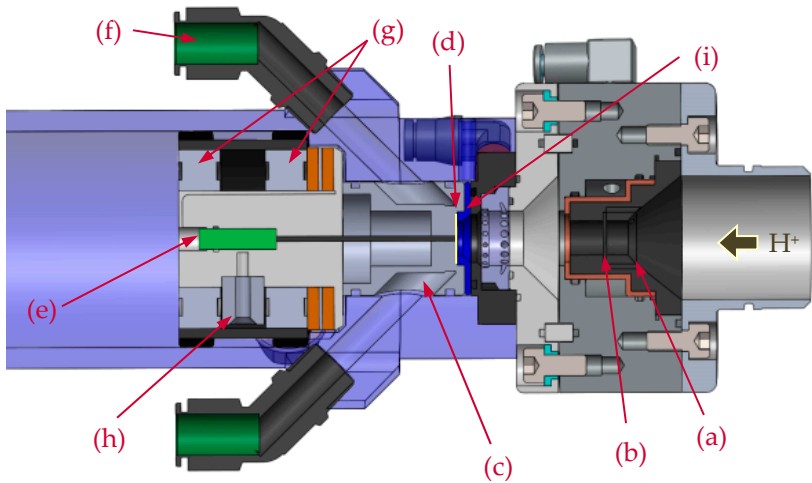

**Figure 1.** Schematic of cast alloy solid target: (a) 10 mm graphite collimator with removable crown; (b) graphite degrader; (c) 3D-printed A357 target holder; (d) target material; (e) type-K thermocouple; (f) 8 mm ID water line;  (g) ceramic bearings; (h) lead ballast; (i) magnetically coupled Nb front cover.

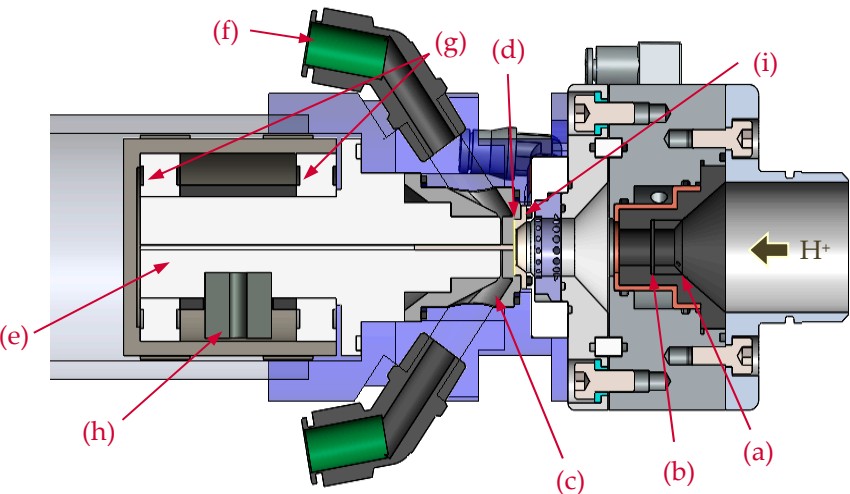

**Figure 2.** Schematic of 3D-printed sterling silver solid target: (a) 10 mm graphite collimator with removable crown; (b) graphite degrader; (c) 3D-printed SS925 target holder; (d) target material; (e) type-K thermocouple; (f) 8 mm ID water line; (g) ceramic bearings; (h) lead ballast; (i) magnetically coupled Nb front cover.

The graphite degrader and collimator are water-cooled to dissipate the heat generated from degrading the cyclotron primary beam energy from 18 to 11.5 MeV. The degrader is interchangeable for specific energy requirements, and is easily accessible from the front by removing the graphite collimator crown.

The new SS925 target system has the same floating design as the previous A357 solid target shuttle [1]. The target body is encapsulated in two ceramic bearings, and free to rotate in the center. A lead ballast is used to self-align the water openings (in the horizontal plane) during the loading process. The complete solution allows the target shuttle to be loaded and unloaded pneumatically from outside the cyclotron bunker. Further enhancement was made to the location of the coolant O-ring seals by placing them on the vertical plane, to assist with the loading and unloading of the target shuttle without resistance, with details shown in Figure 3. The complete target shuttle is pressurized from behind to maintain a tight seal for both water and helium cooling cavities.

Simulations were conducted using SolidWorks 2017/18 CAD package with flow simulation CFD (computational fluid dynamics). The proton beam was modeled as a surface heat source on the target material and assumes that the total power is absorbed on the material surface [1]. The simulation does not account for helium flow on the surface of the target material, since helium cooling is primarily used for the vacuum window only. The CFD package calculated the temperatures for the target material with water applied into the solid target body at an adjusted flow rate of 20 L/min.

The simulated target reflects the same physical dimensions of an electroplated target for the production of $^{64}$Cu. The model target is composed of a Ni layer (100 μm, dia. 7.0 mm) on top of an Au foil (125 μm, dia. 15 mm). The theoretical maximum temperature was calculated for target currents from 40 to 120 μA (at 11.5 MeV) in 10 μA increments. Results were compared for both cast alloy (A357) and sterling silver (SS925) solid target designs.

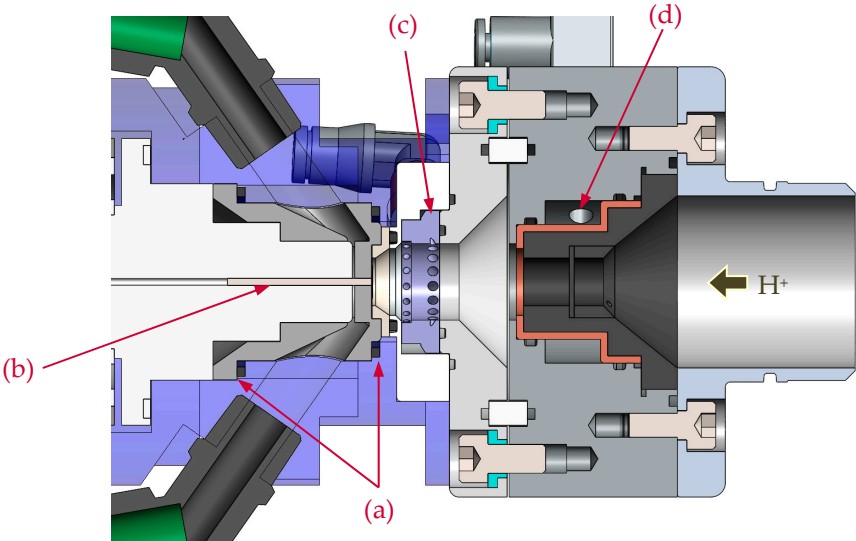

**Figure 3.** Details of 3D-printed sterling silver target (SS925) showing cooling and O-ring seals: (a) O-ring seals; (b) type-K thermocouple; (c) He cooling for target material and vacuum window; (d) water jacket for graphite collimator and degrader.

The solid target shuttle has an embedded Type-K thermocouple (dia. 1.5 mm), located in the center and mounted laterally to the beam path (refer to Figures 2 and 3). The thermocouple is in direct contact with the target material for improved sensitivity and to minimize the surface area exposed to the proton beam. The integration of a thermocouple inside the target holder provides continuous temperature monitoring of the target material during irradiation. Temperature measurements for the 3D-printed target were conducted using a single 125 μm pure Au foil irradiated at 40 μA up to 80 μA, in increments of 10 μA. The sterling silver target and irradiated Au foil were visually inspected for thermal damage for target currents >80 μA.

The 3D-printed sterling silver solid target was validated with the production of radio-copper using enriched $^{64}$Ni electroplated on 125 μm Au foil. This was irradiated at 40 μA to 80 μA for 5 minutes at increments of 10 μA. The maximum temperature was recorded for each increment and compared to the theoretical results from the simulation.

## 3. Results

The advantage of the lost-wax casting process over additive metal 3D-printing techniques is the low porosity of the final metallic form, with a relative density is equivalent to the silver granules used in the casting process, 10.37 g/cm³. As a result, the thermal conductivity is closer to the physical properties of pure silver, 10.49 g/cm³. SLM and other additive 3D-printing processes have moderate porosity with a difficult to determine final relative density. The microstructure from SLM 3D-printing is highly dependent on the scanning speed, hatch spacing, laser power, layer thickness, and the properties of the powder. The relative density for SLM 3D-printed cast alloy can vary from 77% to 97.5% [3], thereby influencing the thermal capability of the target holder. The main disadvantage of the lost-wax casting process is the lower resolution, with wider minimum gap size and greater consideration for escapement holes for the casting material. The surface finish for SS925 is superior to A357, as shown in Figures 4a and 4b. The internal structures are well defined for SS925, with low obstruction to water flow compared to A357. SLM and other additive manufacturing processes require additional post-fabrication machining and polishing. Since the internal cooling structure is inaccessible once the 3D production is complete, the lost-wax casting process is superior for this design and application.

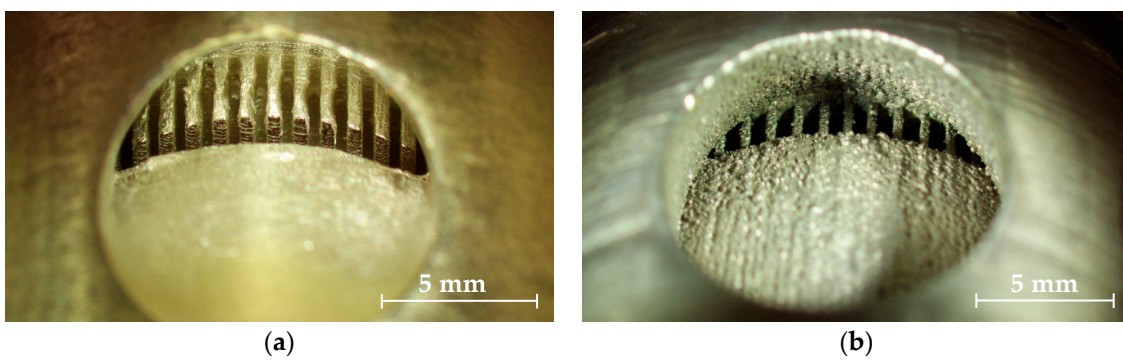

(**a**)                                        (**b**)

**Figure 4.** Internal structure of the water cooling cavity. (**a**) SS925 solid target holder; (**b**) A357 solid target holder.

The maximum theoretical temperatures calculated for the A357 and SS925 solid target stations are shown in Table 1 below. As expected, given the higher thermal conductivity of sterling silver compared to cast alloy (A357), the calculated temperature for SS925 is significantly lower at all target currents.

**Table 1.** Maximum calculated temperatures for A357 and SS925 solid target stations.

| Target Current (μA) | Power (W) | Cast Alloy (A357) (°C) | Sterling Silver (SS925) (°C) |
|---|---|---|---|
| 40 | 460 | 121 | 94 |
| 50 | 575 | 147 | 112 |
| 60 | 690 | 172 | 130 |
| 70 | 805 | 197 | 147 |
| 80 | 920 | 221 | 164 |
| 90 | 1035 | 246 | 182 |
| 100 | 1150 | 270 | 198 |
| 110 | 1265 | 294 | 214 |
| 120 | 1380 | 318 | 231 |

The maximum temperature calculated by the CFD package is observed between the interface of the target material and the surface heat source. In CFD, the thermal advantage of a 3D-printed SS925 target is ~30% compared to A357, and ~60% better than the existing compact solid target system [1]. The simulated temperatures are below the melting points of all relevant target materials, such as Ni ($T_m$ = 1455 °C), Au ($T_m$ = 1064 °C), Y ($T_m$ = 1526 °C), and sterling silver ($T_m$ = 893 °C).

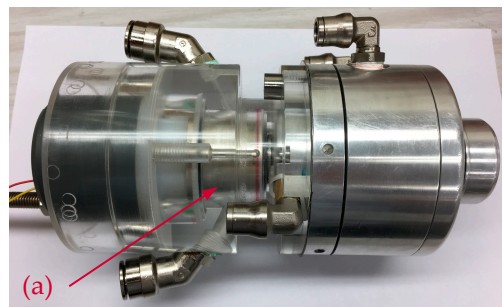

**Figure 5.** Solid target assembly with target shuttle and SS925 target holder (a).

The SS925 solid target system was manually loaded for testing without a PLC-controlled pneumatic loading system. Figure 5 shows the SS925 solid target with the pneumatic shuttle.

Table 2 shows the maximum temperature recorded during the irradiation of the Au foil (without electrodeposited layer) on the SS925 solid target station at various target currents.

**Table 2.** Maximum recorded temperature and visual inspections at 80 µA.

| Target Current (µA) | Solid Target Max. Temp (°C) | Visual Inspection, Damage. (Yes/No) |
|:---:|:---:|:---:|
| 0 | 16 | - |
| 40 | 73 | - |
| 50 | 86 | - |
| 60 | 99 | - |
| 70 | 113 | - |
| 80 | 122 | No |

Visual inspection of the Au foil post-irradiation showed no visible signs of damage, Figure 6b. Beam losses due to divergence are reduced by placing the target material closer to the exit port of the cyclotron. The average beam loss for this design is <25%, compared to >50% for the existing compact solid target on the end of a 30 cm beamline.

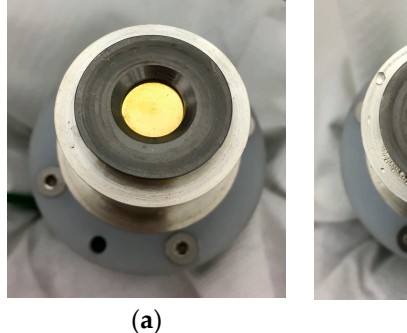 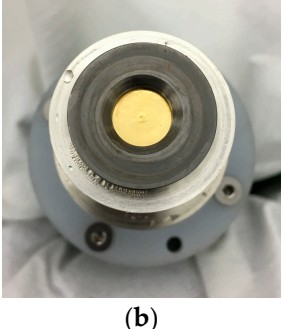

(**a**) (**b**)

**Figure 6.** (**a**) Au foil before irradiation on the 3D-printed SS925 solid target holder. (**b**) Au foil after being subjected to experimental protocol as described above.

A small lump was observed at the center of the Au foil, which indicates the protrusion of the thermocouple from the SS925 body. This reduces the surface contact between the Au foil and the target body, therefore affecting the cooling efficiency.

The maximum temperatures observed during the irradiation of $^{64}$Ni plated on Au for target SS925 is shown in Table 3 below. Once this maximum had been established and the target current stabilized, the temperature remained at a plateau with little variation. The calculated peak temperature is shown in column 3 of Table 3, for comparison with these experimental results. By contrast, from previous experimental results with the A357 target holder [1], the temperature of the assembly would fluctuate erratically if the limit of the material physical property was reached.

**Table 3.** Maximum recorded temperature for $^{64}$Ni on Au using the new solid target system.

| Target Current (μA) | Max. Recorded Solid Target Temp (°C) | Theoretical Temp. (°C) [1] |
|---|---|---|
| 0 | 16 | 18 |
| 40 | 64 | 87 |
| 50 | 74 | 103 |
| 60 | 86 | 119 |
| 70 | 102 | 135 |
| 80 | 118 | 151 |

[1] A thickness of 100 μm of Ni material is used in the theoretical model.

The total mass of the target (enriched $^{64}$Ni and Au) was 478.3 mg with 16 mg of plated enriched $^{64}$Ni. The equivalent thickness of plated $^{64}$Ni was approximately 46.7 μm. The total mass of the target, post-irradiations, was 478.2 mg, which indicates no loss of enriched material from proton bombardment at 80 μA. Figure 7 shows the electroplated enriched $^{64}$Ni on Au pre- and post-irradiation, mounted on the SS925 target holder.

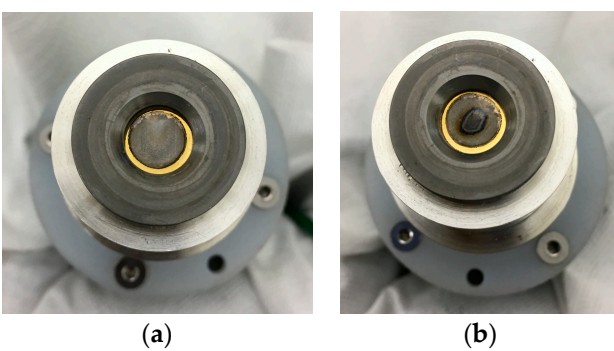

(**a**)  (**b**)

**Figure 7.** (**a**) $^{64}$Ni on Au before irradiation of the 3D-printed SS925 solid target holder; (**b**) $^{64}$Ni on Au after all irradiations as described above, and shown in Table 3.

As determined from SRIM [3], ~1.12 MeV of the degraded primary beam is deposited in the Ni layer, ~5.15 MeV in the Au backing, and the residual 5.23 MeV beam is deposited in the target holder. A thickness of ~100 μm of Ag and stainless steel will effectively stop the residual beam completely. Proton interaction with the thermocouple junction is negligible, since the thermocouple material is encased in a stainless steel sheath with thickness >100 μm. Radionuclide production of $^{106m}$Ag ($T_{1/2}$ = 8.28 d) and $^{107}$Cd ($T_{1/2}$ = 6.5 h), from the 5.23 MeV residual beam, has low excitation functions of <5 mb and <30 mb, respectively [5]. Proton activation of the Ag target holder can be eliminated if the thickness of target material or the gold backing is increased by ~100 μm.

We observed no thermal damage of the Au foil at 80 μA, with no loss of electroplated $^{64}$Ni material. This indicates an improved thermal management of the target current due to the higher thermal conductivity of the SS925 material. The irradiated $^{64}$Ni material appeared tarnished from

proton bombardment in Figure 7b, and is visually similar to a routine 40 μA $^{64}$Cu production on the existing solid target station. The imprint of the beam is clearly visible on the target surface, indicating a sharp approximately Gaussian profile.

Given the assumptions used in the theoretical model for proton bombardment as a homogeneous surface heat source, and the difference in the thickness of the actual enriched $^{64}$Ni layer used experimentally, some differences between theoretical temperatures and measured temperatures for the new solid target system are to be expected. An adjusted simulation model to reflect the true target thickness (46.7 μm) shows that the theoretical model overestimates the temperature by ~35%. The margin of error between true and calculated temperature is indicative of the poor contact between the target material and the SS925 target holder. In addition, the thermocouple transverses the water cavity with active cooling of stainless steel sheath, which may affect the final temperature reading. Given the temperature measurement is used as a guide for target integrity only, the ability to continuously monitor the temperature during an irradiation is invaluable, despite the error. Further refinement to the design, with insulation of the thermocouple from the water channel, is currently in progress. Calibration with destructive testing and thicker Au backing will likely increase the accuracy of the measured temperature and validate the minimal direct influence of the proton beam on thermocouple measurement.

## 4. Conclusions

Theoretically and experimentally, the new solid target design shows significant improvement in temperature management compared to the existing compact solid target system. The new SS925 solid target system offers a more robust and long-term solution with the convenience of loading and unloading the target material outside of the cyclotron bunker. It also moves the target material closer to the cyclotron exit port, thus reducing the beam losses due to divergence along an external beam line. The system is capable of higher target currents without the need to slant the target material to the incident proton beam. This ensures a moderate starting material cost, given the comparatively small amount [4] of enriched $^{64}$Ni material needed for electroplating, compared with traditional slanted targets. Other advantages are the minimal engineering impact to peripheral systems used in the pre- and post-irradiated target preparation and separation processes.

Mounting the thermocouple in the lateral position protects the tip from the incident proton beam and provides direct contact with the target backing. The thermocouple is an effective tool for both beam alignment and live feedback used to monitor the target material during an irradiation. Minor adjustment to the length is needed to minimize its protrusion from the flat surface. This will ensure better target material contact with the SS925 target holder.

Further investigation is needed to validate the temperature observed during a standard irradiation (80 μAh) and to complete the shuttle loading and unloading mechanism. Once the operating temperature profile of the new design is confirmed, it can be adapted to the existing solid target system for greater thermal management at higher target currents.

The SS925 target holder is comparatively economical and efficient, once the design has satisfied all 3D-printing fabrication requirements. Future work will aim to confirm that the 3D-printed SS925 target holder has relatively low activation from proton bombardment at beam energies <11.5 MeV [5], and to compare the yields produced for both $^{64}$Cu and $^{89}$Zr, between our existing compact solid target and the new SS925 solid target system.

**Author Contributions:** Conceptualization, S.C. and D.C.; Methodology, S.C.; Software simulations, S.C.; Fabrication, D.C.; Validation, S.C. and D.C.; Investigation, S.C. and D.C; Resources, R.I.P; Writing—original draft preparation, S.C.; Writing—review and editing, S.C. and R.I.P; Supervision, R.I.P.

**Acknowledgments:** The technical and scientific support of the Radiopharmaceutical Production and Development Laboratory (RAPID, MTP) staff is gratefully acknowledged.

**Conflicts of Interest:** authors declare no conflict of interest.

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
