# Peer review of "Enhancement and Validation of a 3D-Printed Solid Target Holder at a Cyclotron Facility in Perth, Australia"

_instruments, doi:10.3390/instruments3010012_

Reviewer 1 Report

This is an excellent paper from a group that is not afraid to run their pony hard. A few points arise that could be helped with a bit more explanation:  The "target material", namely the Au foil plated with the Ni-64, is pressed against the new silver cooling fixture only by a Nb hold-down clamp, rather than being directly cooled  by a water jet directed onto the rear surface of the Au foil. This (indirect) cooling, while avoiding any possibility of disastrous water leaks, is critically dependent on clamping pressure and the absence of any buckling of Au foil away from the Ag heat sink, which itself depends on stresses arising from the thermal expansion coefficient-matching of the target components. Others have resolved this tendency for thermal runaway by interposing an indium foil between the target material and the cold finger, but this runs the risk of forming corrosive eutectics. The choice of 46 um Ni-64 (delta-E = 1 MeV) and 125 um Au (delta-E = 4.2 MeV) leaves the proton beam with about 6-7 MeV entering the silver cold finger. First, this would seem to directly irradiate the centered K-type thermocouple. How this does not interfere with the temperature readout needs a sentence of explanation. (Perhaps this is the intent of line 169 of the Conclusions.) Also, the activation of the silver, even at 7 MeV, is not negligible, and could be avoided entirely if the Au foil were simply a bit thicker. The most peculiar mention is line 139, stating that SRIM predicts that 22% of the beam is deposited on the Ni-layer. This comment could be less confusing by specifically stating that the 22% refers to the ENERGY LOSS in the Ni-64 relative to the Au backing. A few simple questions: What is the degrader material? 80 uA x 7 MeV = 560 watts; how is it cooled? What is the point of the helium cooling? If it is needed to cool an exit foil protecting the cyclotron vacuum, then the He exit port would provide an alternative temperature sampling station well out of the beam. What is the role of the ceramic bearings? Does the Ag target puck rotate in the beam?   This is a well-written paper addressing real problems as the beam current is pushed upward. These minor revisions would dispel any hesitation on this reviewer’s part.

Author Response

Dear Reviewer,

Thank you for the constructive comments.

Please find here my response to your specific questions.

1. The "target material", namely the Au foil plated with the Ni-64, is pressed against the new silver cooling fixture only by a Nb hold-down clamp, rather than being directly cooled by a water jet directed onto the rear surface of the Au foil. This (indirect) cooling, while avoiding any possibility of disastrous water leaks, is critically dependent on clamping pressure and the absence of any buckling of Au foil away from the Ag heat sink, which itself depends on stresses arising from the thermal expansion coefficient-matching of the target components. Others have resolved this tendency for thermal runaway by interposing an indium foil between the target material and the cold finger, but this runs the risk of forming corrosive eutectics. 

The target material is held in placed by a magnetically coupled Nb ring and does not reply on the clamping pressure to maintain a tight seal for the water cavity. Although the thermocouple transvers through the water cavity to make contact with the target backing, it is a tight fit with some sealant on the thermocouple shaft. During simulations the stresses on the target body was examined to avoid any buckling of the target body due to water pressure and heat. Experimentally we observe no deformation from proton bombardment at > 80µA.

The following lines (74-75) were added to explain the tightness for the target cooling mediums.

The complete target shuttle is pressurized from behind to maintain a tight seal for both water and helium cooling cavities.

2. The choice of 46 um Ni-64 (delta-E = 1 MeV) and 125 um Au (delta-E = 4.2 MeV) leaves the proton beam with about 6-7 MeV entering the silver cold finger. First, this would seem to directly irradiate the centered K-type thermocouple. How this does not interfere with the temperature readout needs a sentence of explanation. (Perhaps this is the intent of line 169 of the Conclusions.) Also, the activation of the silver, even at 7 MeV, is not negligible, and could be avoided entirely if the Au foil were simply a bit thicker?

To address your comments above lines 172-176 was added: 

As determined from SRIM [3], ~1.12 MeV of the degraded primary beam is deposited in the Ni layer, ~5.15 MeV in the Au backing and the residual 5.23 MeV beam is deposited in the target holder. A thickness of ~100 µm of Ag and stainless steel will effectively stop the residual beam completely. Protons interaction with the thermocouple junction is negligible, since the thermocouple material is encased in a stainless steel sheath with thickness > 100 µm.

3. Also, the activation of the silver, even at 7 MeV, is not negligible, and could be avoided entirely if the Au foil were simply a bit thicker?

To address your comments above lines 176-179 was added, include reviewer’s advice: 

Radionuclide production of 106mAg (T1/2 = 8.28 d) and 107Cd (T1/2 = 6.5 h) from 5.23 MeV residual beam has low excitation functions of < 5 mb and < 30 mb, respectively [5]. Proton activation of the Ag target holder can be eliminated if the thickness of target material or the gold backing is increased by ~100 µm.

4. The most peculiar mention is line 139, stating that SRIM predicts that 22% of the beam is deposited on the Ni-layer. This comment could be less confusing by specifically stating that the 22% refers to the ENERGY LOSS in the Ni-64 relative to the Au backing?

To address your comments above the sentence was restructure and rewritten for clarification lines 172-173 as per the response above. 

5. What is the degrader material? 80 uA x 7 MeV = 560 watts; how is it cooled? What is the point of the helium cooling? If it is needed to cool an exit foil protecting the cyclotron vacuum, then the He exit port would provide an alternative temperature sampling station well out of the beam?

To address your comments above lines 52-53 and 64-67 was added: 

The mounting flange for the solid target incorporates a graphite collimator and graphite degrader (dia. 10 mm, thickness 1 mm).

The graphite degrader and collimator are water cooled to dissipate the heat generated from degrading the cyclotron primary beam energy from 18 MeV to 11.5 MeV. The degrader is interchangeable for specific energy requirements and is easily accessible from the front by removing the graphite collimator crown.

-       The size of the schematics was increased with an additional reference to the graphite degrader and collimator as describe above.

6. What is the point of the helium cooling? If it is needed to cool an exit foil protecting the cyclotron vacuum, then the He exit port would provide an alternative temperature sampling station well out of the beam?

To address your comments above lines 78-80 was added: 

The simulation does not account for helium flow on the surface of the target material since helium cooling is primarily used for the vacuum window only.

-       The helium cooling exit port can provide an alternative means to measure the temperature. However, the sensitive for target temperature is diminished as this relies on the thermal conductivity of several layers of different materials relative to the target material. CNRS, France is exploring the use of a pyrometer for target temperature measurement. This non-contact technique has its challenges with vital electronics readers located inside the bunker.    

7. What is the role of the ceramic bearings? Does the Ag target puck rotate in the beam?

To address your comments above lines 68-71 was rewritten:

The new SS925 target system has the same floating design as the previous A357 solid target shuttle [1]. The target body is encapsulated in two ceramic bearings and free to rotate in the center. A lead ballast is used to self-align the water openings (in the horizontal plane) during the loading process.

-       The silver target does not rotate in beam once the target is pressurized from behind to secure the water cooling gallery. We acknowledge the benefit of rotating the target holder and material to homogenise the beam thus reducing the damage on the target material from hot spots. Beam rastering is a well-established technique reserve for large facility with sophisticated beam-line, we can only keep dreaming.

Thank you.

Sun

Reviewer 2 Report

Review of ‘Enhancement and Validation of a 3D-Printed Solid Target Holder at Cyclotron Gacility in Perth, AUSTRALIA’ ID 428402

This manuscript describes a truly innovative new development in medical cyclotron targetry. The paper is well written with all the necessary information present, comparison to a CFD simulation and some experimental results. Some small comments and suggestions:

Page 1: In the description of the 3D printing of A357, is it possible to include the achieved density of the target holder?

Page 2: In the description of the 3D-printing with the lost-wax method of SS975, how much rework or polishing if any was necessary? When jewelry is manufactured with this method, substantial surface work is required after the casting. It would be worthwhile to have a comment here about this.

Figures 1, 2 and 3: The fine details are hard to see in the printed version. Is it possible to make these bigger, e.g. the width of the page? It is especially hard to see the differences between Fig. 2 and 3.

Page 2: In the description of the CFD simulation, where was the maximum temperature measured? Anywhere or like the experiment at the back of the target plate?

Page 4, line 119: I do not understand the sentence ‘The average target-to-collimator ratio is ~75%, compared to < 50% for the existing compact solid target’. Can you please elaborate? Is this about the beam divergence mentioned in the conclusion?

Page 6, line 164. The electroplating cell is not mentioned in the paper. Would it be worthwhile to introduce it earlier on? Here, it seems like an after though.

Author Response

Dear Reviewer,

Thank you for the constructive comments.

Please find here my response to your specific questions.

1. Page 1: In the description of the 3D printing of A357, is it possible to include the achieved density of the target holder?

To address your comments above lines 107-114 was added: 

The advantage of the lost-wax casting process over additive metal 3D-printing techniques is the low porosity of the final metallic form, with relative density equivalent to the silver granules used in the casting process (10.37 g/cm3). As a result the thermal conductivity is closer to the physical properties of pure silver (10.49 g/cm3). SLM and other additive 3D-printing processes have moderate porosity with difficult to determine final relative density. The microstructure from SLM 3D-printing is highly dependent on the scanning speed, hatch spacing, laser power, layer thickness and the properties of the powder. The relative density for SLM 3D-printed cast alloy can vary from 77 % to 97.5 % [3], therefore influencing the thermal capability of the target holder.

 - Additional reference was included.

2. Page 2: In the description of the 3D-printing with the lost-wax method of SS975, how much rework or polishing if any was necessary? When jewelry is manufactured with this method, substantial surface work is required after the casting. It would be worthwhile to have a comment here about this.

To address your comments above lines 47-50 was added: 

The 3D-printed holders had minimal post-fabrication treatment with light polishing of accessible surface only. In-house machining of the 3D-printed part includes O-ring grooves, thermocouple hole and polishing the interface where the target material is located.

- Additional figures to compare the surface finish was included

Also, lines 116 – 121 was added

The surface finish for SS925 is superior to A357 as shown in Figures 4a and 4b. The internal structures are well defined for SS925, with low obstruction to water flow compared to A357. SLM and other additive manufacturing process require additional post fabrication machining and polishing. Since the internal cooling structure is inaccessible once the 3D-production is complete, the lost wax casting process is superior for this design and application.

3. Figures 1, 2 and 3: The fine details are hard to see in the printed version. Is it possible to make these bigger, e.g. the width of the page? It is especially hard to see the differences between Fig. 2 and 3.

Noted, Done – thank you for the advice.

4. Page 2: In the description of the CFD simulation, where was the maximum temperature measured? Anywhere or like the experiment at the back of the target plate?

To address your comments above lines 131-132 was added: 

The maximum temperature calculated by the CFD package is observed between the interface of the target material and the surface heat source.

Page 4, line 119: I do not understand the sentence ‘The average target-to-collimator ratio is ~75%, compared to < 50% for the existing compact solid target’. Can you please elaborate? Is this about the beam divergence mentioned in the conclusion?

To address your comments above the sentence was rewritten (lines 146-148): 

Beam losses due to divergence are reduced by placing the target material closer to the exit port of the cyclotron. The average beam loss for this design is < 25 %, compared to > 50 % for the existing compact solid target on the end of a 30 cm beamline.

Page 6, line 164. The electroplating cell is not mentioned in the paper. Would it be worthwhile to introduce it earlier on? Here, it seems like an after though.

To address your comments above the sentence was rewritten (lines 209-210):

Other advantages are the minimal engineering impact to peripheral systems used in the pre-and post-irradiated target preparation and separation processes.

Thank you

Sun

Reviewer 3 Report

This is an interesting study on 3D printing of a solid target for production of PET radiometals.  The paper is well organized but the English language and grammar needs to be revised throughout the manuscript.  This represents an important development for solid target design.

Specific comments:

What enterprise did the "commercial" 3D printing of the sterling silver target?

Since the irradiations at each level of beam current were only for 10 min, can you comment on whether temperature was rising during the 10 min intervals or had it come to a plateau?

Author Response

Dear Reviewer,

Thank you for the constructive comments.

Please find here my response to your specific questions.

1. What enterprise did the "commercial" 3D printing of the sterling silver target?

To address your comments above, line 36- 37:

The sterling silver (SS925) solid target holder was ordered online via the Shapeways 3D-printing manufacturing portal Shapeways

-      Due to commercial reasons, Shapeways do not disclose its supplier. However previous consultation with the manufacturer due to the complexity of the design we know the origin of the manufacturer is in New Jersey, USA.  

2. Since the irradiations at each level of beam current were only for 10 min, can you comment on  whether temperature was rising during the 10 min intervals or had it come to a plateau?

To address your comments above, line 155-160 was added: 

The maximum temperatures observed during the irradiation of 64Ni plated on Au for target SS925 is shown in Table 3 below. Once this maximum had been established and the target current stabilized, the temperature remained at a plateau with little variation. The calculated peak temperature is shown in column 3 of Table 3, for comparison with theses experimental results. In contrast, from previous experimental results with the A357 target holder [1] the temperature would fluctuate erratically if the limit of the material physical property was reached. 

Extensive review of the English language was conducted in response to the reviewer’s concern.

Thank you.

Sun

Round  2

Reviewer 1 Report

My previous review comments were completely resolved by the author's responses. This is a remarkable paper, finally unleashing 3-D printing to bear on a real problem in our field. I congratulate this group in pushing the beam-current limits forward, an essential advance in providing radio-metals to a wider user group.

Reviewer 2 Report

Dear Authors,

The rework of the manuscript was very successful and I recommend to accept the paper in its present form.